# More than Just a Number: Perspectives from Black Male Participants on Community-Based Interventions and Clinical Trials to Address Cardiovascular Health Disparities

**DOI:** 10.3390/ijerph21040449

**Published:** 2024-04-06

**Authors:** Faith E. Metlock, Sarah Addison, Alicia McKoy, Yesol Yang, Aarhea Hope, Joshua J. Joseph, Jing Zhang, Amaris Williams, Darrell M. Gray, John Gregory, Timiya S. Nolan

**Affiliations:** 1Johns Hopkins School of Nursing (Formerly The Ohio State University College of Nursing), Baltimore, MD 21205, USA; fmetloc1@jhu.edu; 2Washington University School of Medicine (Formerly The Ohio State University College of Medicine), St. Louis, MO 63110, USA; saddison@wustl.edu; 3OhioHealth (Formerly The Ohio State University Center for Cancer Health Equity), Columbus, OH 43202, USA; alicia.mckoy@ohiohealth.com; 4The Ohio State University Comprehensive Cancer Center, Columbus, OH 43210, USA; yang.6310@osu.edu (Y.Y.); jing.zhang@osumc.edu (J.Z.); 5Nell Hodgson Woodruff School of Nursing (Formerly The Ohio State University College of Nursing), Atlanta, GA 30322, USA; aarhea.hope@emory.edu; 6The Ohio State University College of Medicine, Columbus, OH 43210, USA; joshua.joseph@osumc.edu (J.J.J.); amaris.williams@osumc.edu (A.W.); 7Gray Area Strategies LLC (Formerly The Ohio State University College of Medicine), Columbus, OH 43210, USA; dgrayiimd@gmail.com; 8The African American Male Wellness Agency, National Center for Urban Solutions, Columbus, OH 43205, USA; jgregory@ncusolutions.com; 9University of Alabama at Birmingham Heersink School of Medicine (Formerly The Ohio State University College of Nursing and The Ohio State University Comprehensive Cancer Center), Birmingham, AL 35233, USA

**Keywords:** black men, clinical trial participation, medical mistrust, community-based participatory research, health equity

## Abstract

Background: Black Americans remain significantly underrepresented and understudied in research. Community-based interventions have been increasingly recognized as an effective model for reckoning with clinical trial participation challenges amongst underrepresented groups, yet a paucity of studies implement this approach. The present study sought to gain insight into Black male participants’ perception of clinical trials before and after participating in a community-based team lifestyle intervention in the United States. Methods: Black Impact, a 24-week community-based lifestyle intervention, applied the American Heart Association’s Life’s Simple 7 (LS7) framework to assess changes in the cardiovascular health of seventy-four Black male participants partaking in weekly team-based physical activities and LS7-themed education and having their social needs addressed. A subset of twenty participants completed an exit survey via one of three semi-structured focus groups aimed at understanding the feasibility of interventions, including their perceptions of participating in clinical trials. Data were transcribed verbatim and analyzed using a content analysis, which involved systematically identifying, coding, categorizing, and interpreting the primary patterns of the data. Results: The participants reported a positive change in their perceptions of clinical trials based on their experience with a community-based lifestyle intervention. Three prominent themes regarding their perceptions of clinical trials prior to the intervention were as follows: (1) History of medical abuse; (2) Lack of diversity amongst research teams and participants; and (3) A positive experience with racially concordant research teams. Three themes noted to influence changes in their perception of clinical trials based on their participation in Black Impact were as follows: (1) Building trust with the research team; (2) Increasing awareness about clinical trials; and (3) Motivating participation through community engagement efforts. Conclusions: Improved perceptions of participating in clinical trials were achieved after participation in a community-based intervention. This intervention may provide a framework by which to facilitate clinical trial participation among Black men, which must be made a priority so that Black men are “more than just a number” and no longer “receiving the short end of the stick”.

## 1. Introduction

Black Americans make up 13% of the US population, but only account for approximately 9% of clinical trial participants, despite federal mandates for the inclusion of racial and ethnic minorities and women in clinical trials [1]. This is troubling, especially considering that Black Americans are at an increased risk for chronic health conditions including diabetes, cancer, and cardiovascular disease (CVD) when compared to their White counterparts [2]. In particular, Black American men suffer from an earlier onset of [3] and higher mortality rates [4] for CVD, and have the shortest life expectancy compared to other non-indigenous race/sex groups in the US [5,6]. A number of factors influence life expectancy in Black men, including socioeconomic barriers [7], perceived racism [8], social networks [8,9], health knowledge or awareness [8], and masculinity beliefs [10,11]. As clinical trials provide novel and life-changing therapies, increasing participation among Black American men is vital [12,13,14]. Without adequate representation, it becomes difficult to ensure generalizable findings, due to a potential lack of variability in responses in homogenous populations [15]. A notable example is the documented differential response observed in Black patients in hypertension management, for which calcium channel blockers are more effective than angiotensin-converting enzyme inhibitors as monotherapy [16]. Clinical trial participation among those who are at the highest risk of chronic disease, like Black American men, remains an ever-present concern for progress toward health equity [14].

Persistent challenges with engaging Black Americans extend from clinical research through healthcare utilization [8,17,18]. Dispelling the unfounded myth that Black Americans are less inclined to participate in research [19,20], a more plausible explanation points to the responsibility of investigators. This is evident in the analysis of 100 cardiovascular clinical trials funded by the National Institutes of Health (NIH), revealing a prevalent absence of intentional enrollment goals for Black patients, often resulting in unmet or undisclosed targets [21]. The repercussions of these actions extend to even more harmful and unjust downstream effects, leading to missed opportunities to address critical community-identified needs, comprehend responses to medical therapies, and create data-driven solutions for enhancing the delivery of high-quality cardiovascular care to those most in need [14].

Addressing the pronounced underrepresentation of Black adults in NIH-funded cardiovascular trials, coupled with the lack of specific enrollment plans, calls for targeted interventions that thoroughly tackle barriers and facilitators across every phase in a given study. Identified barriers to the participation of Black Americans in clinical trials include medical mistrust, concerns of integrity among researchers, a lack of information, time and resource constraints, opportunity costs, as well as racism and discrimination [13,14,22,23]. Unsurprisingly, many of these barriers can be traced back to the long and sordid history of medical abuse and experimentation on Black Americans [24]. However, to increase the representation of Black Americans in clinical trials, researchers must aim not only to alleviate barriers, but also encourage facilitators of Black Americans’ participation. In contrast to the extensive knowledge of barriers affecting clinical trial participation, there is a noticeable dearth of information on facilitating factors, particularly among Black American men. The existing literature does highlight some facilitating factors, such as the use of research ambassadors, the employment of racially and ethnically diverse staff, and investments in communities. However, it is important to note that only a few studies have employed the focus group methodology—a proven effective approach in unveiling beliefs, perceptions, and attitudes among racial minority groups—to glean information on facilitating factors for clinical trial participation [25].

On the continuum of community-engaged research, community-based participatory research (CBPR) has increasingly been recognized as an effective model for reckoning with clinical trial participation challenges amongst underrepresented groups; yet, there is a paucity of studies implementing CBPR approaches [26]. Even fewer studies focus on improving CVD in Black American men, which is concerning given the aforementioned disparities in CVD mortality [27]. Focused on co-learning and capacity building among all partners, CBPR harnesses the distinctive strengths of academic and community partners through a framework that (1) balances research and action for the mutual benefit of all partners, (2) utilizes the unique strengths of academic and community partners, and (3) promotes collaborative and equitable partnerships across all research phases to empower all and share power [28]. By engaging stakeholders throughout the entire research process, CBPR contributes to the development of sustainable solutions for the community and advocates for health policy changes that target and mitigate health disparities [29]. The current study explores perceptions of clinical trials and clinical trial participation among Black American men in a 24-week community-based clinical trial of a cardiovascular health intervention (based on the American Heart Association’s Life’s Simple 7; LS7) [9].

## 2. Methods

### 2.1. Study Design

The parent study examined the feasibility and preliminary efficacy of Black Impact, a 24-week community-based lifestyle intervention aimed to improve LS7 among Black men, and has been described in detail previously (ClinicalTrials.gov Identifier: NCT04787978; Ohio State University Institutional Review Board Number: 2019H0302) [9]. Briefly, in central Ohio, the study team as well as community organizations and members adapted the Diabetes Prevention Program [30,31] and American Heart Association Check, Change, Control programs, applying evidence-based strategies and stakeholder feedback. Each participant was assigned to one of six teams which included 8–25 participants who lived in close proximity to a central meeting location (e.g., Columbus Recreation and Parks recreation center). Within teams, men received 45 min of physical activity led by a personal trainer and 45 min of health education led by health coaches weekly. Participants were not randomized in the single-arm trial and partook in the entire intervention.

### 2.2. Data Collection

Using a convenience sampling technique, three months after completion of the Black Impact community–based lifestyle intervention, all participants (74 Black men) were invited via email by study staff members (TSN and AM) to participate in exit surveys conducted exclusively through focus group interviews. Men that accepted the invitation received an additional focus group consent form, which was reviewed at the beginning of the interview sessions with verbal consent to participate. The data collection involved three semi-structured, virtual focus groups facilitated by a single interviewer (TSN) with two notetakers (AM and FM). Each focus group had five to ten participants. The interview protocol guided discussions to ascertain information relating to their experiences within the Black Impact study, as well as their perceptions of health, social support, and clinical trial participation. Relative to the present study, the interview protocol prompted men to recall their perceptions of clinical trials before and after participating in the Black Impact study with special attention to factors identified as barriers and facilitators of participation in found in the literature. (Appendix A). All responses from the focus groups were transcribed verbatim and reviewed for accuracy. 

### 2.3. Analysis

The qualitative analysis was grounded in an inductive approach, allowing themes to emerge directly from the rich narratives of participants, thus enabling a deeper understanding of clinical trial participation in the context of Black men. Content analysis was utilized to systematically explore the dataset and discern overarching patterns and nuanced commonalities. Raw field notes and verbatim transcripts, embodying the multifaceted reality of participants’ experiences, constituted the initial data corpus. This process involved the development of a manageable classification or coding scheme, known as an a priori code list, derived from the interview protocol. Drawing upon both participant-generated constructions and analyst-generated constructions [32], four coders (TSN (a qualitative methods expert), YY, FEM, and JZ) independently analyzed each transcript for significant statements, applying a logical framework to identify emergent patterns in the data. Coders meticulously categorized participants’ statements into specific codes aligned with larger descriptors, facilitating the identification of overarching themes. Through iterative meetings, codes were refined and consolidated until achieving 100% intercoder agreement, allowing for additions or subtractions of codes as necessary. This collaborative process of theme construction and refinement enabled the identification of salient insights and nuanced interpretations. Furthermore, the dataset was thoroughly examined, and data saturation was attained. Integral to finalizing themes was the active engagement of a community partner (JG) with 20 years of experience in designing and implementing directed programs (e.g., biometric screening, mental wellness chats, financial education) aimed at improving health among Black men, underscoring the commitment to community-based participatory research principles. The final themes were shared with this partner for validation, ensuring alignment with the lived experiences and perspectives of the community. This manuscript details themes relative to participants’ perceptions of clinical trial participation after participating in a community-based intervention. 

### 2.4. Positionality Statement

We recognize that the perspectives and biases drawn from our diverse team’s backgrounds in academia, healthcare, and community advocacy have shaped our approach to this study. Throughout the research process, we have remained mindful of our positions of privilege and power, acknowledging our influence on data interpretation. Our collaboration with a community partner was motivated by our commitment to centering the voices of the men. The title of the manuscript, a direct quote from the men, reflects our intent to allow their voices to resonate. While our interpretations are situated within our own perspectives, we endeavor to offer insights into the complexities of clinical trial participation. Our goal is to foster a nuanced understanding of and support for community-based interventions addressing these issues. In presenting our findings, we strive for critical engagement to deepen our understanding of healthcare disparities and community-based interventions rooted in methodological choices that prioritize cultural relevance and ongoing community collaboration to foster trust and rapport. 

## 3. Results

Twenty Black men with a mean age of 52 years participated in the focus groups. A total of 35% of the focus group participants were married, 25% were divorced, and 35% were never married. Their income ranged from <USD 20,000 (6%) to ≥USD 75,000 (24%). A total of 80% were employed, 15% were retired, and 5% were unemployed. A total of 70% of the participants were enrolled in private insurance, 15% in Medicaid or Medicare, and 5% in military insurance. A total of 10% of the participants were uninsured (Table 1).

Themes co-constructed from the data were relevant to their perceptions of participation in clinical trials prior to and after their participation in the community-based clinical trial (Table 2).

### 3.1. Pre-Community-Based Clinical Trial Participation

The participants shared similar, apprehensive thoughts on clinical trials prior to participating in the community-based study. Three main themes contributed to their perception of clinical trials: (1) History of medical abuse; (2) Lack of diversity amongst research teams and participants; and (3) Positive view of racially concordant research teams.

History of medical abuse. When asked about their thoughts towards clinical trial participation prior to this study, their recollection of historical events of medical abuse in research, for example, the United States Public Health Syphilis Study at Tuskegee [33], commonly provoked uneasiness towards participating in clinical trials for the men. One man echoed: “Historically, as far as African Americans and clinical trials, we’ve got a bad history and a lot of mistrust and distrust with it”. Another participant supported the importance of having a racially concordant research team considering past abuse: “I’m looking at all the doctors and I want to see okay this is a group that is really looking with Black people in mind. We need to have studies done by us, for us that aren’t what we’ve historically, in the past, been subjected to”.

Lack of diversity amongst research teams and participants. The lack of diversity in research teams and participants also notably influenced the men’s reluctance to participate in clinical trials prior to this study. A participant expressed: “Most of the participants from our understanding were usually not people of color, so again how those certain trials and treatments would work on people of color may be different in how they would work on someone who was not a person of color”. Another participant compared a previous research experience to our study: “Other than the coordinator, there was no one else you know who looked like me and I just didn’t get the same type of energy from participating so it was more of a challenge and wasn’t a good experience ultimately. This time around was much better”.

Positive view of racially concordant research teams. When asked “how did the inclusion of Black health coaches influence your willingness to participate, if at all?”, most of the men noted that having a racially concordant research team strongly influenced their decision to participate in a clinical trial. One participant reflected: “I’ve been fortunate that the ones that I’ve done have been led by or had a team that have majority Black folks… I know this is a group that is really looking with Black people in mind, so I don’t mind participating”. Another participant similarly expressed his desire for a racially concordant team: “There has to be research done on Black folks by Black folks because I was hesitant about many things in the past”. A participant expresses the benefit of racial concordance, “I believe that there are many things in the African American and Brown communities that someone that looks like them understands what they’re going through, understands our diet throughout the history of the African American community, it can be more beneficial to one’s health”.

### 3.2. Post-Community-Based Clinical Trial Participation

Overall, most of the men conveyed a positive outlook on clinical trials after participating in the community-based study. The three main themes that influenced their change in attitude towards clinical trials were as follows: (1) Building trust with the research team; (2) Increasing awareness about clinical trials; and (3) Motivating participation through community engagement efforts.

Building trust with the research team. Many of the men mentioned that racial concordance with the research team and study participants increased their level of trust when participating in the clinical trial. A participant mentioned: “I never had that many young Black urban professionals that tended to my personal needs. If you guys knew how that made me feel, I was the one so proud. I am a Pan African to the utmost, and like I said [it’s] just a beautiful thing to see young people so involved”. Another participant supported racial concordance, saying “when you have Black researchers actually speaking up on our behalf and coming to us with the information, it makes a major difference”. A few participants also noted the research team’s delivery of the clinical trial information promoted their sense of trust: “With you all being the ones who were bringing this information to us, it was pretty clear and direct. I didn’t feel like it was any hidden agendas, the only agenda, I thought was to increase the health of Black men and, I thought that was good, so I didn’t have any apprehension”. This trust extended outside of the study, given that the men spoke about attributing more value to the findings stated by the study team’s providers. One man in particular reported feeling skeptical when presented test results from a non-racially concordant provider, but after discussions with the study team’s providers, the participant gave the finding more credence, stating the information provided by the study team is more tailored to a “Black” perspective. While it is not entirely clear from where this increased sense of trust with our team’s providers stemmed, the participants emphasized that the development of personal relationships with the research team was a factor in fostering a sense of trust. A participant reflected: “I can confidently speak for everybody on the call and pretty much everybody in the program. We never felt like we were just a number. I always felt like a human person and so that connection was authentic, and I think that’s why we were so responsive and continued to be engaged because we didn’t feel like a number”.

Increasing awareness about clinical trials. The participants mentioned that the clear and direct delivery of the purposes of this study, how it would be executed, and the importance of clinical trial participation supported their decision to participate in the trial. A participant reflected: “You were very straightforward, you said exactly what it was about. Black men need to take care of themselves, and this study was to find out how Black men… could change by doing this, doing that and I like how that came about”. Other participants shared similar narratives. One participant explained: “I was able to be on board with the Black Impact because of the way it was presented at the luncheon to where they literally broke down the whole program. We had an understanding of what to look forward to versus some of these clinical trials where they are just you know say read this, take this, then we see what happens. I’m not down with that, I think I’m still in guinea pig status, but something like this program, I can get with”. The participants highlighted that learning about the negative implications influenced by the paucity of research on Black participants increased their desire to participate in the clinical trial: “There is now intentionality in making sure that Black people are a part of the some of the research that are being done, and he actually gave us some information on how we could be a part of that” stated a participant. Another participant shared how the study’s clear focus on people of color was encouraging: “To say that this particular trial is being focused on the health and wellness of people of color, I’m certainly happy to be a part of their research and happy to be the beneficiary of the research”.

Motivating participation through community engagement efforts. When asked “what drew you into participating in this study?”, the participants mentioned that it was relatively simple to find motivation to participate in the clinical trial due to their health consciousness and desire to improve their health. “I knew that I needed to do something… it wasn’t a real hard decision to decide to participate”, said a participant. Of note, the men viewed a community-based lifestyle change intervention as a safe alternative compared to participating in a trial testing the effect of a drug. A participant said: “I’ve never seen or heard of a clinical trial in this regard, where you know it wasn’t taking anything orally, so this was a positive experience but again I’m not really up, for you know trying medications and chance they work or whatnot”. Another factor introduced by participants was inclusion of community engagement efforts as a means of support for recruitment and retention. One participant said, “If a church brings someone in that wants to do research, then then likelihood of them getting participants would be a little bit higher than just trying to get it from the general public”. A participant also shared how the study team influenced their motivation: “the doctors, as a team that was just motivating for me, I was like you know, here we have a team that really cares about us. I’m going to do all I can to get the best out of this”. Most of the men expressed that the brotherhood created amongst the participants most notably motivated their continued participation in the trial. A participant reflected: “I don’t know what would’ve kept me in besides the brotherhood”. When asked by the interviewer “how, if at all, did participation in the study change your view on clinical trials”, the men shared similar sentiments that participation in the Black Impact study positively changed their perceptions towards participating in future studies. “I’m more open to possible participating in the future, whereas before you know, I have no desire whatsoever or never thought about it, but I look at it differently now”, a participant shared. Another participant stated, “he suggested researchmatch.com and I am a participant with that, and I’ve been through a couple of research surveys and the whole nine, so it had me more willing participant in those things”.

## 4. Discussion

It has been nearly 30 years since the 1993 NIH Revitalization Act, for which researchers were tasked with increasing representations of Black, indigenous, and other people of color and women in clinical trials [34]. Since that time, some progress has been made on uncovering the etiologies behind low clinical trial participation, particularly that of Black Americans. However, this understanding has not led to a significant increase in the participation of Black Americans. This fact is amplified in cardiovascular disease drug trials, in which Black Americans have about a 60% prevalence of total cardiovascular disease [5] but are still inadequately represented in cardiovascular clinical trials [35,36]. When examining this reality further, sex differences exist: Black women account for 6% of clinical trial participation in cardiovascular disease trials compared to 3% for Black men [36,37]. By treating the men as more than “just a number” through building trust with research teams, increasing awareness about clinical trials, and motivating participation through community engagement efforts, Black Impact was able to accrue and retain Black men in a clinical trial.

The present study sheds light on the factors that encouraged the Black men who participated in a community-based lifestyle intervention to think differently and more positively about clinical trial participation. Initially, most participants (even those who had participated in clinical trials previously) expressed apprehensive sentiments regarding clinical trial participation prior to the study. This perception was expectedly influenced by their knowledge on the history of medical abuse by researchers on Black Americans, a lack of diversity amongst research teams and participants, and a positive perception of racially concordant research teams. We are still challenged with the catastrophic effects that studies like the US Public Health Syphilis Study at Tuskegee [33] and the Henrietta Lacks [38] story imposed on Black Americans’ willingness to participate in clinical trials. Moreover, the lack of diversity seen in most research teams and studies further complicates Black Americans’ eagerness to participate. Understanding that there will likely be apprehension amongst this group in recruitment, researchers must acknowledge the vile history of medical abuse inflicted on Black Americans and build trust, increase awareness of clinical trials through outreach initiatives, and motivate participation through community engagement efforts. After participating in this study, in which we thoughtfully incorporated these elements, most of the men noted a positive change in their perception of clinical trial participation. This sentiment resonates with findings from a previous study which reported that African Americans compared to White Americans who participated in a clinical trial are similarly likely to participate in future research studies [39]. Notably, there was no disaggregation of sex reported in the study. Noting the aforementioned stark differences of clinical trial participation among Black men and women, more disaggregated research is needed to elucidate perceptions of and willingness to participate in clinical trials.

As noted, building trust with the research team can be accomplished through relationship building. The research team of the Black Impact study capitalized on its existing partnership with a trusted community organization to build initial interest in Black Impact, a tactic that has been successful in other settings [40,41]. Engaging key stakeholders was a critical component of community engagement, given that focusing solely on clinical trial recruitment without any input or cooperation from community members may have proven less successful [42]. Though the partnership was beneficial, many Black men queried reported that interpersonal relationships between the other men participating in the trial and members of the research team built their confidence to participate in the study [43]. For several others, this rapport spoke to the benevolent intent of the research team, which likely had an essential role in reducing the mistrust associated with clinical trials among Black men. Relationships, both individual, interpersonal, and community-wide may provide a veritable “North Star” to guide future clinical trial engagement strategies [44].

The present study reveals that Black men identify racial concordance with the research team as a positive factor influencing their involvement in clinical trials. The participants spoke favorably about seeing a research team composed of individuals who looked like them. The available evidence either supporting or refuting the effectiveness of concordance between the research team and participants is scant [45], yet this observation aligns with the existing literature on provider–patient communication during clinical encounters [46]. While racial concordance alone is not enough to quell decades of mistrust built through historical instances and amplified by personal experiences, racial concordance can be a key component of the initial trust built between the participants and research staff. Another study found that racial concordance between researchers and participants could soften Black Americans’ initial trepidation towards clinical trial participation [22]. In fact, the vast majority of Black men interviewed in this study highlighted that having a racially concordant research team strongly influenced their participation. Though it is unlikely that every clinical trial research team can be entirely racially concordant with Black participants, the intention to engage and communicate in a culturally humble manner that acknowledges an individual’s perspectives and shares power may be an avenue to overcome participant apprehension [47]. To better facilitate this, researchers may consider the addition of concordant “patient navigators” or “community health workers” to their teams [48]. In a 2016 study examining the retention of Black Americans in cancer clinical trials, researchers found that the use of these navigators, who explain the trial in clear detail, keep participants updated, and customize support measures for them, increased retention among Black Americans [49].

Regarding increasing awareness about clinical trials, many of the participants credit the study team for their transparency throughout the research process. Lack of understanding about clinical trials has been identified as a barrier among racial and ethnic minorities [23,50]. A qualitative study with both urban and rural participants similarly discovered that improving their knowledge about clinical trials served as a motivator to participate [51]. Our team intentionally sought to increase understanding by clearly explaining all aspects of the study to the participants and highlighting its importance through well-organized presentations. Due to instances of mistreatment of Black Americans by the healthcare system, the transparency exhibited by the research team was critical to building trusting relationships. This heightened level of transparency may also have influenced the expectation held by some Black Americans that they are not as highly valued by physicians and researchers as White Americans [22,52]. A similar issue affecting the poor recruitment of Black Americans is neglecting to inform these individuals of clinical trials [50]. Once informed, many of the participants of the Black Impact study described themselves as easily encouraged to join trials [39,53]. Such observations underscore the essential nature of community partnerships to access and provide necessary information about clinical trial participation to Black men. 

With regard to motivating participation through community engagement efforts, the type of clinical trial offered may affect one’s willingness to participate. Within this study, some participants mentioned a greater willingness to participate in community-based interventions like Black Impact rather than experimental drug trials. Though little evidence exists exploring this phenomenon in Black Americans, historical abuse inflicted upon Black Americans may undergird this sentiment. A recent study found that when participants were given the chance to choose between two hypothetical trials, they were more likely to partake in an observational diagnostic trial than a drug trial [54]. However, examining this relationship among Black Americans represents an area of future of inquiry. 

Another rationale discovered for low Black clinical trial participation is the notion that clinical trials are not conducted for people of color and/or the results will not benefit racial and ethnic minority groups [22]. As a participant stated, a way to mitigate concerns that clinical trials may not be conducted for people of color is to meaningfully focus on targeted recruitment within the Black community, a strategy made more attainable through partnerships with local community organizations [12,44,55]. Moreover, educating participants on the lasting impact their research participation may have in advancing health knowledge among Black Americans may be an effective way to motivate Black participation. A similar qualitative study observed a theme related to the “value of research” and found that African American participants expressed a greater inclination to participate in research if they were made aware that the disease was prevalent in the African American community [25]. Therefore, researchers must make it clear to participants that other Black people may benefit from their clinical trial participation. Furthermore, study findings showed that Black men appreciated being included and valued in clinical trials, which is another strategy for motivating participation. Rather than perpetuating the traditional separation that exists between participants and researchers, a team-based collective framework instilled a sense that the study participants were members of the research “team” [23]. These sentiments were achieved by community-engaged efforts including continuous transparency, the personal investment of researchers and staff alike, and the consideration of the participants as not only patients but as more than “just a number”.

It is incumbent upon researchers to increase the representation of groups made vulnerable or groups with a higher burden of chronic disease like Black men in clinical trials. The success of the Black Impact study in terms of clinical trial participation can be accredited to the use of community-engaged principles [28]. To promote the inclusion of Black men in clinical trials, it is imperative for researchers to engage this community meaningfully by addressing well-noted barriers such as mistrust and research literacy. In addition to recognizing the importance of racial concordance within the research team, it is equally vital to embrace diversity among team members and participants, acknowledging that inclusivity does not necessitate homogeneity in race, but rather a commitment to diverse perspectives and experiences [56]. Engagement is best supported by academic–community–government–industry partnerships that thoughtfully share decision-making responsibilities, working toward a shared goal [55]. Working with partners in research, as well as its implementation and dissemination, lends itself to designing an optimal research study that effectively recruits and retains groups traditionally believed to be “hard to reach” [12,29]. The subsequent section delineates community-engagement strategies utilized by the Black Impact study throughout all study phases, with a particular focus on addressing barriers and enhancing the recruitment and retention of Black men, offering insights for researchers seeking to enhance diversity in clinical trials.

## 5. Strategies for Applying Community Engagement across Study Phases

### 5.1. Pre-Clinical Trial

#### 5.1.1. Engaging Community Members as Active Participants in the Research Team

Integrating the community as an essential part of the research team goes beyond conventional methods like forming community advisory boards. A recommended approach to gain insight into facilitators for clinical trial participation involves actively incorporating community members as integral contributors to the research team. By fostering a collaborative environment with shared power and resources, the engagement of community members from study design to implementation promotes a sense of ownership and trust. This, in turn, enhances participant recruitment and retention. A systematic review of randomized clinical trials found that only 5% of studies included patient, family, or community participation. Further, the stakeholders’ involvement was mainly in the implementation phase, underscoring the importance of augmenting stakeholder engagement across the entire spectrum of research [57]. The Black Impact study, our community-based clinical trial, stands out as an exemplary model in this regard. We established an intervention working group in collaboration with a trusted community partner, the African American Wellness Agency (AAMWA). Through iterative collaboration with the community organization, we gained invaluable insights into barriers and facilitators for potential participants, deepened our understanding of community-identified problems, explored potential solutions, and obtained valuable information about available resources to mobilize and sustain our efforts.

#### 5.1.2. Evaluating Inclusion and Exclusion Criteria 

To avoid perpetuating disparities among communities made vulnerable, researchers must thoughtfully consider the inclusion and exclusion criteria in clinical trials, striving for equitable access to potentially life-saving therapies. Traditional eligibility criteria often result in the disproportionate exclusion of underrepresented groups, especially those with organ function deficiencies or prevalent comorbidities among minorities [58,59]. Researchers must actively address biases in recruitment sites that may categorize certain individuals as “at risk”, thus influencing their enrollment [58]. Research teams could further benefit from training on cultural humility and contexts related to health inequities [60]. It is important to note that although racial concordance may not be feasible in all studies, researchers can promote inclusivity through the ongoing practice of cultural humility as they prioritize diversity amongst their research teams and participants. The focus should shift from exclusionary practices to adaptable analyses, ensuring findings are disaggregated to elucidate potentially important differences in health risks and outcomes that may be tested in future powered trials.

#### 5.1.3. Employing Tailored Recruitment Strategies

Recruitment strategies that are refined to ensure culturally relevant messaging and outreach resonance with Black men are key drivers of success in recruitment. Through a partnership with the AAMWA, the Black Impact study exemplified the importance of designing recruitment strategies guided by community input. The investigators of Black Impact facilitated recruitment at community events sponsored by the AAMWA, providing an opportunity to directly engage with potential participants. Additionally, the study team conducted educational outreach initiatives to raise awareness about clinical trials and the purpose of the Black Impact study. These relationship-centered recruitment [61] efforts not only provided a forum for participants to pose questions and voice concerns but also afforded researchers the opportunity to address these concerns, cultivating a sense of trust and transparency.

### 5.2. Clinical Trial 

#### 5.2.1. Incorporating Opportunities That Afford Comradery and Fellowship

After participants have been enrolled in the study, promoting camaraderie and fellowship becomes paramount for building trust and encouraging active participation. This imperative was acknowledged and effectively addressed in the Black Impact study through the implementation of strategies aimed at bolstering social support [43]. By organizing teams based on residence, the study encouraged regular interaction among participants, extending beyond the scheduled in-person study sessions. The resulting sense of brotherhood within these teams played a significant role in shaping participants’ engagement and satisfaction with the community-based clinical trial. The study’s emphasis on social support, both from the study team and fellow participants, not only heightened their interest but also contributed to improved adherence to the study protocol and overall participant retention [62].

#### 5.2.2. Addressing Potential Roadblocks to Participation

Recognizing potential roadblocks to participation is vital, particularly when recruiting participants such as Black men, who often contend with competing demands from their work and personal lives. Researchers must move beyond mere recognition and proactively address barriers like transportation, cost, and time constraints to ensure equitable access to clinical research opportunities [60]. The Black Impact study modeled this by strategically scheduling the weekly in-person study sessions for evening hours to encourage attendance. Additionally, the investigators aimed to minimize travel time and enhance accessibility by selecting study sites in recreational parks and community centers located near the participants’ residences. To compensate for study tasks completed outside of the in-person sessions, such as surveys, the participants were provided appropriate financial compensation for their time.

### 5.3. Post-Clinical Trial 

#### 5.3.1. Hosting Community Engagement Forums

Hosting community engagement forums post-clinical trial plays a crucial role in the recruitment and retention of Black men in future research studies. These forums provide an opportunity to gain valuable insights into participants’ impressions, receive feedback, and disseminate key findings. Directly sharing findings with participants ensures that they are active beneficiaries of the research outcomes and aids in educating other members of the community [28]. More importantly, these forums contribute to building and maintaining relationships with former participants, establishing a foundation for ongoing engagement in future opportunities. For Black Impact, this commitment to post-trial engagement was evident through continuous efforts to keep the men involved. The study sought ways to keep the participants engaged through regular updates, subsequent study opportunities such as focus group sessions, and providing ongoing information on the study’s outcomes. This iterative process of engagement, feedback, and sustained communication is vital for not only refining research practices but also fostering a participant-centered approach that extends beyond the duration of the initial clinical trial.

#### 5.3.2. Conducting Long-Term Community Impact Assessments

Long-term community impact assessments, which entail an ongoing and thorough evaluation of sustained effects within a community, have proven to be invaluable, especially when dealing with vulnerable communities frequently exposed to helicopter research. Helicopter research, characterized by brief and superficial observations or interventions without establishing meaningful engagement, often hinders a comprehensive understanding of community dynamics, akin to a helicopter hovering briefly over a site without in-depth integration [63]. This becomes especially crucial in mitigating the negative effects of transient or exploitative research practices. Black Impact recognized this importance and, in collaboration with participants from the study team and the AAMWA, engaged in several meetings to reflect on and learn from study experiences. The insights gained from these meetings were instrumental in informing the development of Black Impact 2, demonstrating a commitment to sustaining a positive community impact and avoiding the pitfalls associated with short-term, disconnected research approaches (Table 3).

As mentioned in the Future of Nursing Report, implementing studies that target disadvantaged groups is critical to advancing health equity [64]. Despite the extensive recognition of barriers to recruiting and retaining Black Americans, minimal progress has been made in terms of increasing the representation of this group in clinical trials. To reach the goal of the National Institutes of Health Minority Health and Health Disparities’ strategic plan to increase the overall proportion of participants from diverse populations included in NIH-funded clinical research to 40% by 2030 [65], researchers must prioritize recruiting and retaining underrepresented groups by thoughtfully creating sustainable studies rooted in community-engaged efforts that address well-documented barriers and utilize facilitators that promote participation [13].

## 6. Limitations

Our results are taken in the context of some important limitations. First, our sample only included participants that opted to participate in the semi-structured, virtual focus groups. Therefore, our findings are not generalizable to all the male participants from the Black Impact Study. Additionally, all the participants in the focus group sessions have prior research participation experience from the lifestyle intervention. Thus, our sample lacks representation from people who have never participated in a clinical trial. However, it is important to note that the exclusion of participants who have never participated in clinical trials aligns with the specific focus of this paper, which is centered on factors promoting clinical trial participation. Including perspectives from those who have not engaged in clinical trials may not contribute meaningfully to our investigation into enhancing participation rates. Lastly, the exit focus groups occurred three months after the completion of the intervention, and the outcomes may have differed if the sessions were conducted closer to the intervention. Despite its limitations, our study reports the sex-disaggregated data of Black men who had previously participated in clinical trials yet still had apprehension towards participating in the cardiovascular community-based clinical trial. The findings from this work support future community-engaged endeavors to apply our lessons learned and improve clinical trial participation among this disparate group of people.

## 7. Conclusions

Black participants, especially Black men, remain underrepresented and understudied in most clinical research studies despite having notable health inequities. Previous studies have indicated barriers and facilitators to clinical trial participation amongst groups made vulnerable; however, few studies have employed the focus group methodology to identify facilitating factors to clinical trial participation through the context of community engagement. This analysis captured the perspectives of Black men on clinical trials and the impact of participating in the Black Impact study, a community-based clinical trial, on their pre-existing assumptions. Presented here are valuable lessons and strategies on the recruitment and retention of Black men in clinical trials, which may lend themselves to the creation of trials with more clinical diversity and support avenues to build health equity. Improved perceptions of clinical trial participation were achieved after participating in the current community-based study. Significant factors that influenced this change in perception included the research team’s conscious efforts to promote trust, increasing the Black men’s awareness of clinical trials, and motivating participation in clinical trials through community-engaged efforts. Black men’s inclusion in clinical trials must be made a priority, so that Black men are “more than just a number” and no longer “receiving the short end of the stick”.

## Figures and Tables

**Table 1 ijerph-21-00449-t001:** Age and sociodemographic characteristics of focus group participants vs. non-focus group participants in Black Impact.

	No Focus Group (N = 54)	Focus Group (N = 20)	Overall (N = 74)	*p*-Value
Age	51.8 (10.5)	53.3 (10.3)	52.2 (10.4)	0.583
Marital Status				0.169
Married	31 (57.4%)	7 (35%)	38 (51.4%)	
Widowed	0 (0%)	1 (5%)	1 (1.4%)	
Divorced	8 (14.8%)	5 (25%)	13 (17.6%)	
Separated	1 (1.9%)	0 (0%)	1 (1.4%)	
Never Married	11 (20.4%)	7 (35%)	18 (24.3%)	
Missing	3 (5.6%)	0 (0%)	3 (4.1%)	
Annual Income				0.682
<USD 20,000	3 (5.6%)	2 (10%)	5 (6.8%)	
USD 20,000–49,999	13 (24.1%)	7 (35%)	20 (27%)	
USD 50,000–74,999	16 (29.6%)	6 (30%)	22 (29.7%)	
≥USD 75,000	13 (24.1%)	3 (15%)	16 (21.6%)	
Missing	9 (16.7%)	2 (10%)	11 (14.9%)	
Employment Status				0.612
Employed	44 (81.5%)	16 (80%)	60 (81.1%)	
Retired	4 (7.4%)	3 (15%)	7 (9.5%)	
Unemployed	4 (7.4%)	1 (5%)	5 (6.8%)	
Missing	2 (3.7%)	0 (0%)	2 (2.7%)	
Health Insurance				0.908
Private insurance	38 (70.4%)	14 (70%)	52 (70.3%)	
Medicaid/Medicare	5 (9.3%)	3 (15%)	8 (10.8%)	
Military insurance	3 (5.6%)	1 (5%)	4 (5.4%)	
No insurance	7 (13%)	2 (10%)	9 (12.2%)	
Missing	1 (1.9%)	0 (0%)	1 (1.4%)	

Legend: numbers are mean (standard deviation) or count (percentage). *p*-values were calculated using *t*-tests or X2 tests, where appropriate.

**Table 2 ijerph-21-00449-t002:** Perceptions of clinical trial participation: exemplar quotes from participants pre- and post-community-based intervention.

		Exemplars
Pre-community-based clinical trial participation	History of medical abuse	“I will say I was hesitant about many things due to the past”
	Lack of diversity amongst research teams and participants	“With this particular trial being focused on the health and wellness of people of color, I’m certainly happy to be a part of their research and happy to be the beneficiary of the research”
	Preference for racially concordant research teams	“Black was major. It is of major importance to me to be Black”.
Post-community-based clinical trial participation	Building trust with the research team	“I’m thankful for the agency connecting me with Ohio State to be a part of something of this caliber. I thought it was a professional presentation of how to engage the community in a positive way and Black men, specifically in a positive way to show that there’s that there is an activity of caring for black men”.
	Increasing awareness about clinical trials	“All things that the study dove into and letting folks know you can’t just look at your physical body, but also what’s going on internally was definitely a positive thing to see happening. Hopefully folks continue to do this because, again, you may feel great, but there might be a lot of things going on inside your body that you just don’t know and you don’t want it to be too late for you to corrective action, so that was encouraging for me”.
	Motivating participation through community engagement efforts	“It was almost like if somebody in the Black Impact program came to me and say hey you have a terminal issue with your health do you want your life saved? Yes!”

**Table 3 ijerph-21-00449-t003:** Strategies for applying community engagement across study phases.

Stage of Research	Strategy
Pre-Clinical Trial	Engaging community members as active participants in the research team
	Evaluating inclusion and exclusion criteria
	Employing tailored recruitment strategies
Clinical Trial	Incorporating opportunities that afford comradery and fellowship
	Addressing potential roadblocks to participation
	Conducting long-term community impact assessments
Post-Clinical Trial	Hosting community engagement forums
	Conducting long-term community impact assessments

## Data Availability

Data are available on request due to restrictions, e.g., privacy or ethical reasons. Because of the sensitive nature of the data collected for this study, access to the data set from researchers trained in human subject confidentiality protocols may be requested from the corresponding author.

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
