# Peer review of "More than Just a Number: Perspectives from Black Male Participants on Community-Based Interventions and Clinical Trials to Address Cardiovascular Health Disparities"

_ijerph, 2024, doi:10.3390/ijerph21040449_

Round 1

Reviewer 1 Report

Comments and Suggestions for Authors

Abstract

Consider adding the United States since this is an international audience. Add more detail on analysis. Currently states "Data were transcribed verbatim and analyzed using content analysis". Add a sentence that describes the content analysis.

For key words - Consider "medical mistrust" over "mistrust"

Introduction- 

More paragraph breaks are needed for ease of reader.

Methods

Line 129 - Should say "convenience sample". It currently says "convince sample". 

For Data Collection - Please include a supplementary material with the interview guide. Also, please specify if they were focus groups or 1:1 exit surveys. Section 2.2 needs some clarification.

In section 2.3, please include more detail on thematic analysis.

Results

Table 1 - Add more detail to title, tables should stand alone.

Table 2 - Add more detail to title, tables should stand alone.

Discussion

More solutions for researchers looking to include more diverse populations in CTs. Please indicate in a sentence or two that the subsequent section focuses on more solutions for researchers looking to include more diverse populations in CTs. Excellent points made in this section.

Conclusion

Consider shortening.

Author Response

Thank you so much for your thoughtful feedback on the manuscript. Please see the attachment which details point-by-point how we addressed the comments you raised. 

Reviewer 2 Report

Comments and Suggestions for Authors

Ethical principles must be included and the measures to ensure trustworthiness. The researchers must briefly describe the conceptual/theoretical framework that may underpins the study. The researcher should describe the research paradigm and justification. The researchers must avoid to used sources older than 10 years. 

Author Response

(The authors gave the same response as above.)
